# Domains of Physical and Mental Workload in Health Work and Unpaid Domestic Work by Gender Division: A Study with Primary Health Care Workers in Brazil

**DOI:** 10.3390/ijerph19169816

**Published:** 2022-08-09

**Authors:** Marta Regina Cezar-Vaz, Daiani Modernel Xavier, Clarice Alves Bonow, Jordana Cezar Vaz, Letícia Silveira Cardoso, Cynthia Fontella Sant’Anna, Valdecir Zavarese da Costa

**Affiliations:** 1School of Nursing, Federal University of Rio Grande, Rio Grande 96203-900, Brazil; 2Faculty of Nursing, Federal University of Pelotas, Pelotas 96010-610, Brazil; 3Institute of Dermatology Professor Rubem David Azulalay (Medical Residency), Rio de Janeiro 20020-020, Brazil; 4Department of Nursing, Federal University of Pampa, Uruguaiana 97501-970, Brazil; 5Department of Nursing, Federal University of Santa Maria, Santa Maria 97105-900, Brazil

**Keywords:** gender, women, men, health worker, mental health, workload, primary health care, unpaid domestic work

## Abstract

Various studies indicate that workload metrics can be used to assess inequities in the division of labor according to gender and in the mental health of health care professionals. In most studies, the workload is portrayed in a way that does not integrate the different fields of work, that is, work in health services and unpaid domestic work. The objective was to determine the effects of the workload domains of health work and unpaid domestic work according to the gender division of health professionals working in primary health care (PHC), and to analyze the workload as an inducer of anxiety disorders and episodes of depression. This cross-sectional study consisted of 342 health care professionals recruited for interview at primary health care units in the extreme south of Rio Grande do Sul, Brazil. Sociodemographic and occupational variables, workload in PHC and unpaid domestic work, and dichotomies of anxiety disorders and episodes of depression were considered. Poisson and multivariate linear regression models were used for data analysis. Cohen’s standardized effect size was used to assess the magnitude of the difference between women and men in terms of workload. The female professionals presented higher scores in terms of PHC work and unpaid domestic work and higher proportions of episodes of depression and anxiety disorders compared to males. The male professionals showed that anxiety disorders presented a medium standardized effect size on domestic workload and the level of frustration with family involvement was higher in those with episodes of depression. The results illustrate that the workload metric is an important indicator of female vulnerability to working conditions in PHC and in the family environment.

## 1. Introduction

Primary health care (PHC), based on the universal concept of the World Health Organization (WHO) [1], is considered as the organizer of the health care network and coordinator of comprehensive care in the Brazilian Unified Health System. In other words, it is the first level of health care and is characterized by a set of health actions at the individual and collective levels, covering the promotion and protection of health, disease prevention, diagnosis, treatment, rehabilitation, harm reduction, and health maintenance to develop comprehensive care, which positively impacts the health of communities. It is the main gateway to the SUS and the communication center with the entire SUS Care Network and should be guided by the principles of universality, accessibility, continuity of care, comprehensive care, accountability, humanization, and equity. This means that PHC works as a filter, which is capable of organizing the flow of services in health networks from the simplest to the most complex. The PHS is structured around public health outpatient facilities (basic health units, UBS) that refer patients to other more complex levels of care [2]. Therefore, PHC develops differentiated care for workers by incorporating the contribution of work in the determination of the health–disease processes [3].

Accordingly, the World Health Organization (WHO), based on the Sustainable Development Goals, advocates for the extension and quality of the coverage of the service in PHC, based on fair and safe work processes without reference to the worker’s gender, which would integrate work and family care into healthy work processes for both sexes [4,5]. The human workforce of PHC professionals, men and women who develop the process of assisting populations in their communities, is included in this broader context. These PHC professionals work intensively to achieve and improve the resolution capacity of the actions developed through the health team. According to the World Health Organization [4], women represent 70% of the health and social care workforce worldwide but occupy only 30% of the leadership positions in the area of health. These macrosocial characteristics establish a scientific and solidary concern regarding the maintenance of studies that can evidence changes or constants in working conditions from the perspective of the division of labor by gender (men and women). Work in the health area is composed of different scenarios of action, with the focus of this study being PHC.

The division of labor by gender can be highlighted, referring to the way in which paid work and unpaid domestic work is divided between men and women according to their gender [6]. It can be affirmed that, even in our current socio-historical moment, the division of labor between men and women takes place in everyday life. These characteristics can result in social and organic vulnerabilities, such as a lower social value of women’s work compared to the social value of men’s work, resulting in salary differences for the same job, that is, better wages for men, jobs for women and not for men, for example, in the area of nursing and social assistance, as previously indicated in a report by the World Health Organization [4], and potential biological problems, such as those related to physical and mental health. This social determinant, work, unfolds in the relationship between paid work, with the work of PHC professionals being of interest here, and unpaid work, focusing on the unpaid domestic work performed by these professionals in the family environment.

For this study, unpaid domestic work is the work performed by adults with their families, comprising household activities, such as cleaning, tidying, cooking, and taking care of clothes, in addition to taking care of children, the elderly, people with disabilities and animals, among other activities that are necessary not only for the reproduction of the workforce, but, more broadly, of life, guaranteeing the well-being of different conformations of families in society. People need to work and generate income to satisfy their economic needs (paid work, as in the specific case of health professionals) and also need to take care of their families, which means that unpaid domestic work must be performed [7,8].

The valorization of the conditions of the division of labor according to gender should consider both men and women [6], aiming to prevent diseases and injuries. Accordingly, technological resource metrics should be used that can be evaluated through the workload. The workload metric has different concepts, for example, any physiological effort resulting from reactions [9], or the function of the number of hours worked [10,11]. For the purposes of this study, the definition of Hart and Staveland [12] and Hart, 2006 [13] was adopted, in which the workload is a multidimensional subjective construct, defined by the perception of the effort expended to reach a certain level of performance. Following on from this author’s perspective, the concept stands out against the reality of the world of work, since organizations expect that the activities required in different jobs are carried out quickly, accurately, and reliably using the available resources. As this is often not possible, especially when trying to force the best performance, the work process can generate human costs (fatigue, stress, illness, and accident). Managers and workers need answers, making the concept of professional workload a valuable metric for knowledge of the human cost of performing tasks with high productivity (Hart 2006) [13].

These are defined as a dynamic balance between the demands of the task and the individual’s response when evaluating the perceived workload, to prevent the emergence of burnout, fatigue, decreased performance, frustration, increased risk of accidents and errors, with serious consequences for the performance of work activities [12,13,14]. In addition, they strengthen knowledge, rights and singular and collective gender needs for health promotion and surveillance at work, since human pathological fatigue cannot be ignored in favor of high productivity in production processes.

From the perspective of the division of labor according to gender, health systems can use this metric to study work in PHC. A study carried out in Spain, which evaluated the workload of doctors in a primary health unit in the Health District of Granada-Metropolitana, highlighted the significantly higher workload during the COVID-19 pandemic compared to prior to the pandemic (66.1% versus 48.6% before COVID-19). This increase was greater for women at the physical, temporal and frustration levels [14].

Considering occupational exposure according to PHC workers’ gender in relation to the workload characteristics of physical, mental and temporal demands, performance and frustration, unpaid domestic work, professional activity, time in the profession and sociodemographic profile, it was found that the workload has a greater influence on women regarding the health care work process. Gender studies carried out on health care professionals in the United Kingdom, Canada, Austria and Brazil have shown that the feminization of the global health workforce presents a challenge for human resource policies. This is because, in the areas of nursing, medicine, dentistry and physiotherapy, the workforce is largely composed of married women with little professional experience, and entails long working hours [5,15,16].

Epidemiological studies carried out with health care professionals found that women are at greater risk for the development of workload-related diseases, especially musculoskeletal [17] and mental disorders [18,19]. These are usually the result of the significant average weekly working hours and unfair division of labor in domestic marital relationships, which results in an “invisible” workload being imposed on women who take care of the house and children [20].

The mental burden in PHC is shown to be a determinant of the multifaceted mental disorders of public health work, which are generally related to women’s work and depend on the type of work activity and working conditions, combined with the unpaid domestic work [21].

According to the International Statistical Classification of Diseases and Related Health Problems (ICD-10), mental disorders include illnesses such as depression, bipolar affective disorder, schizophrenia, anxiety disorders, stress, substance abuse disorders, sleep–wake cycle, mental deficiencies and developmental and behavioral disorders [22]. Accordingly, gender studies with health care professionals show that the women’s mental workload is usually greater, leading to work-related stress. This is associated with their excessive care for the home and family, which often culminates in anxiety disorders and episodes of depression, affecting the effectiveness and efficacy of patient care work, as well as family involvement [23,24].

Inadequate working conditions, combined with the PHC work activities, can add to the workload. In Brazil, work in this type of care is developed in a decentralized manner and takes place in the location closest to the people’s home. There are several related government strategies, one of which is the Family Health Strategy (Estratégia de Saúde da Família—ESF), which provides multidisciplinary services to communities through, for example, Family Health Units (Unidades de Saúde da Família—USF). In these, activities such as the reception, consultations, exams, vaccines, radiographs and other procedures are made available to users [2]. From a gender perspective, the intense flow of paid work, added to the family responsibilities of unpaid domestic work, usually causes work strain in women, with anxiety disorders and episodes of depression resulting from this process.

Furthermore, it should be emphasized that PHC workers deal with risks in their paid work activities. A scoping review carried out in Australia showed that health care professionals are at potential risk of harm, due to exposure to various hazardous agents found in their workplace. These include biological risks, mainly due to the occupational transmission of blood-borne pathogens, such as hepatitis B, HIV and hepatitis C, through needle/sharp injuries and splash accidents; psychosocial risks, such as workplace violence, burnout and job dissatisfaction; ergonomic risks, such as musculoskeletal pain, and chemical risks, such as exposure to latex [25]. From the perspective of gender, women in PHC perform their paid working activities in conditions that require qualitative and quantitative improvements related to the effort made by the health care professionals. In this way, the study of workload from the perspective of bias in the division of labor by gender in PHC is relevant, since it contributes to the knowledge about the health conditions of workers in the health profession, in which women constitute about 70% of the workforce in the workspaces [4].

Another factor that contributes to the PHC workload refers to unpaid domestic work. A gender study with health care professionals in Spain during COVID-19 found that female health care professionals who had two or more children were more susceptible to stress, depression and anxiety, compared to those who had one or no children, with increased family responsibilities, an unequal division of work at home and exhaustion, contributing to women’s workload with family involvement [26].

Thus, studies in the literature about the workload of PHC workers and those in unpaid domestic work were shown separately, mostly in paid work, and not related to the division of labor by gender. In this sense, it is worth highlighting the present study, which integrates the study of workload in two work environments, PHC work, and unpaid domestic work, from the perspective of the division of labor by gender. Given this discovery, comprehensive measures and public health actions are imperative, which require adjustments in the model of care for workers’ health in PHC, with local and global political decisions that assess the accumulation of workload. The workload generated in these two work environments can affect health professionals; that is, the workload in PHC can affect unpaid domestic work due to their socially complex nature, which contributes to work and family conflict, illness, and unproductivity at work.

Accordingly, this study aimed to determine the effects of workload domains (mental demand, physical demand, temporal demand, performance, total effort, frustration and overall/total workload) on health work and unpaid domestic work, according to the gender division of PHC professionals, as well as analyzing the workload as an inducer of anxiety disorders and episodes of depression in health care professionals, according to division of gender in work in PHC and unpaid domestic work (family involvement).

## 2. Materials and Methods

The present cross-sectional study investigated the relationship between workload in PHC work and unpaid domestic work (family involvement) using a self-report, based on a medical diagnosis, of the presence or absence of anxiety disorders and episodes of depression among female and male health care professionals. This study was carried out in two municipalities in the extreme south of the state of Rio Grande do Sul, Brazil. The sample size was calculated using the StatCalc tool of Epi Info^®^ (version 7.2, CDC, Atlanta, GA, USA). A total of 548 workers working in PHC during the study period were considered, including nurses, doctors, dentists, nursing technicians, community health agents and oral health assistants/technicians, with a margin of error of 5%, confidence level of 95% and losses of 5%. Professionals were selected through non-probabilistic sampling, with an intentional consecutive sample that contained a minimum of 232 professionals from the area covered by the study. The inclusion criterion was a history of working in PHC for at least six months. The exclusion criterion was being off work for any reason during the data collection period, from January to March 2020.

The sample consisted of 342 professionals, including 53 nurses, 43 doctors, 73 nursing technicians, 139 community health agents, 13 dentists, 15 oral health technicians/assistants and 6 others. The others correspond to professionals who worked in PHC in a few health teams (1 physical therapist, 1 physical educator, 2 psychologists and 2 nutritionists) and were included in the study to consider the characteristics of these health teams. A total of 14 interviews were lost, due to the interview being scheduled and not carried out after three attempts, and 5 professionals did not agree to participate in the research, making a total of 19 professionals that were not interviewed, 3 male workers and 16 female workers. It is noteworthy that most research participants were female health professionals. This numerical differential regarding the gender of the professionals is based on the context of this study and data from the IBGE [27], which show that 78.9% of the workforce in primary health care is made up of female professionals. This information allows for us to conclude that the predominance of female workers in the sample reproduces the more general context of work in PHC in Brazil.

Considering the representativeness of each job category and the complexity and diversity of the actions developed by workers in outpatient primary health care, it was decided that the sample should be composed up to the last moment of participation in the data collection period. This a priori decision was anchored in the perspective of similarities and differences in the work, which were carefully addressed in the data analysis. The attributions of the professionals in primary health care teams followed the aforementioned legal provisions that regulate the performance of each of the professions [28]. Community health agents (CHAs) are professionals who fulfill the role of mediators between technical and common knowledge and between the health team and the community, as well as strengthening the population’s access to health services [29].

From January to March 2020, male and female health care professionals were personally recruited at their workplaces by previously trained researchers. These researchers were always in pairs or trios to ensure their safety and speed up the selection process. It is notable that, in this period, the education and research activities of the university had not yet been suspended in the face of the COVID-19 pandemic.

For data collection, individual and face-to-face interviews with workers from primary health care (PHC) outpatient facilities were used, with an average duration of 58 min. A structured questionnaire was used, including sociodemographic data (age, self-reported skin color, marital status, education, number of children), work in PHC (location of work, other work parallel to PHC, type of primary health unit, profession, length of professional experience, time working in PHC, total number of hours worked in the week, working hours in PHC, monthly income), and the presence/absence of anxiety disorders and episodes of depression. To assess the subjective workload in PHC work and unpaid domestic work, the National Aeronautics and Space Administration (NASA) Task Load Index (TLX) (NASA-TLX) [12,13] scale was applied, as detailed in this section.

The variables of the structured questionnaire (sociodemographic and occupational variables, PHC workload and home workload, and dichotomous variables on anxiety disorders and episodes of depression) were tested and adjusted in meetings in the Laboratory for the Study of Socioenvironmental Processes and Collective Health Production (LAMSA) and through a pilot study, prior to data collection, with a sample of ten individuals (women and men) from different health-care-provider categories. The main propositions of this previous study aimed to evaluate and adapt the data collection instrument regarding the effectiveness of its application and the cognitive understanding of the participants in relation to their ease or difficulty responding to the questions, as well as to improve the qualification of the field researchers. An outline of the key concepts and operational qualifiers used in this study to investigate the magnitude of the difference between problems for female and male professionals in primary health care (PHC) is found in Figure 1.

Mental disorders were assessed through the PHC professionals’ self-reports of medical diagnoses that met the definitions of the Ministry of Health [29] and the ICD-10 according to the World Health Organization (WHO) [22], which are as follows: generalized anxiety disorder (F41.1), with generalized and persistent anxiety but not restricted to, or even strongly predominant in, any particular environmental circumstances (i.e., “free-floating”); dominant symptoms are variable, but include complaints of persistent nervousness, tremors, muscle tension, sweating, dizziness, palpitations, dizziness, and epigastric discomfort. Depressive episodes (F32), where the patient suffers from lowered mood and reductions in energy, activity, interest and concentration; sleep is often disturbed and appetite diminished; ideas of guilt or worthlessness are often present.

To identify the workload in PHC work and unpaid domestic work, the NASA-TLX scale was applied. This was designed to capture the subjective experience of workers [12,13]. This scale has been applied in different work environments [30,31,32,33,34] and was validated in terms of its conceptual and operational structures in Brazilian studies [27,28]. Furthermore, the research group had used this scale in previous studies [35,36,37,38,39,40]. These studies prove that the operator/worker-centric NASA-TLX scale has consistency, sensitivity, and acceptance among operators. Additionally, as it is a multidimensional scale, it can be applied to obtain more detailed and diagnostic data that question what produces the workload reported by workers and can point to strategies for relieving excessive workload.

This scale has the following six domains/dimensions for measuring workload: 1. mental demand (mental and perceptual activity required by the task, such as thinking, deciding, calculating, remembering, looking, searching, etc.); 2. physical demand (physical activity required by the task, such as walking, pushing, pulling, turning, sliding, controlling, etc.); 3. temporal demand (time required and whether the pace of work is slow or fast); 4. performance (how successful the worker believes they were in performing the activities that their work requires); 5. total effort (mental and physical effort needed for the worker to maintain their level of performance); 6. frustration (feelings of insecurity, discouragement, and irritation caused by the work) [13]. Health care workers were asked to assign a score from 0 to 20 (0 = no demand and 20 = maximum demand) to their workload according to the six previously mentioned domains [13]. Following NASA-TLX procedures, workload assessment results using this scale are classified into four workload levels as follows: 0–5, low; 6–10, medium–low; 11–15, medium–high; 16–20, high. (Figure A1 in Appendix B). The NASA-TLX scale was applied to health care workers twice, with one application corresponding to their experience in primary health care work and another to their domestic work with their family. The reliability of these variables was assessed through Cronbach’s alpha, which was 0.805 for the domestic workload (unpaid domestic work) and 0.767 for the workload in primary health care (PHC) work.

Statistical analysis was performed using SPSS v. 21.0 (IBM Corp., Armonk, NY, USA). Quantitative variables were described through mean and standard deviation or median and interquartile range. Categorical variables were described through absolute and relative frequencies. To compare means, Student’s *t*-test for independent samples was applied. In cases of asymmetry, the Mann–Whitney test was applied. This nonparametric test replaced the Student’s *t*-test when the variable did not present a normal distribution [41]. Cohen’s standardized effect size (SES) was used to assess the magnitude of the difference between women and men in terms of workload. According to Cohen (1988) [42], an effect size below 0.5 is considered small, an effect between 0.5 and 0.79 is moderate and an effect equal to or above 0.8 standard deviations represents a large effect. The level of significance was 5% (*p* ≤ 0.05).

To assess the associations between the qualitative variables, Pearson’s chi-square or Fisher’s exact tests were used. To control for confounding factors, multivariate linear regression analysis was used for the quantitative outcomes. The regression or angular coefficient (*b*) was calculated, which measures the effect on the outcome of each one-unit increase in the factor, along with the 95% confidence interval (Appendix A). The criterion for entering the variable in the model was that it had a *p* value of <0.10 in the bivariate analysis with gender. For the categorical outcomes, a Poisson regression model was used (Appendix A). The prevalence ratio was calculated, together with the 95% confidence interval, to measure the independent effect of gender (men and women) on the results.

All participants were informed of the study objectives and signed two copies of the consent form prior to participating in the research. The study was conducted in accordance with the Declaration of Helsinki, and the protocol was approved by the Research Ethics Committee (Conep) of the Federal University of Rio Grande (CAAE: 70043717.0.0000.5324).

## 3. Results

A comparison of the sociodemographic and work-related variables according to gender is presented in Table 1. The sample consisted of 342 health care professionals with a mean age of 41.5 years (±10.0), predominantly women (*n* = 297; 86.8%). When comparing male and female professionals, female professionals were significantly older (*p* = 0.033), were less educated (specifically, more incomplete/completed high school; *p* = 0.002), had more children (*p* = 0.002), lower incomes (specifically less than two minimum wages; *p* < 0.001), had a lower proportion of other work in parallel with the PHC (*p* < 0.001), were more often nurses and community health agents (*p* < 0.001), had spent more time in the profession (*p* = 0.011), had spent more time in PHC work (*p* < 0.001) and had completed fewer total weekly working hours (*p* = 0.033).

The mean workload scores for primary health care (PHC) work and unpaid domestic work, according to the gender of the health care professionals (men and women), are shown in Table 2 and Figure A2 and Figure A3 (Appendix B). The minimum and maximum value of these scores ranged from 0 to 20. In primary health care (PHC) work, it was found that the women had a high mean workload in the domains of mental demand and total effort, which, together with physical demand, frustration and overall/total workload, were significantly higher in women than in men.

When assessing the unpaid domestic work, the women also presented with a significantly higher workload than the men in the domains of physical demand, total effort, frustration and overall/total workload. Furthermore, considering the workload domains in both work environments, the standardized effect size (Cohen’s SES) for the significant differences ranged from 0.32 to 0.68, that is, from small to medium effect by gender. This was more pronounced in the frustration domain.

After adjusting for possible confounding factors (Table 2), the gender (male and female) of the health care professionals was independently associated with mental demand (*p* = 0.013), physical demand (*p* = 0.029), frustration (*p* < 0.001) and total workload (*p* = 0.020) for the PHC work and frustration levels for unpaid domestic work (*p* = 0.003).

In primary health care (PHC) work, female health care professionals presented, on average, an increase of 2.26, 2.22, 4.18 and 1.62 points in the levels of mental demand, physical demand, frustration and total workload, respectively, when compared to men. In addition, in unpaid domestic work, female professionals presented a mean increase of 3.32 points in frustration levels compared to male professionals.

An evaluation of the workload according to the presence or absence of anxiety disorders according to the gender of the professionals is shown in Table 3 and Figure A4 (Appendix B). In female health care professionals who self-reported anxiety disorders, the workload domains (mental demand, temporal demand, frustration, and total workload) in both work environments, that is, in PHC work and unpaid domestic work, were significantly higher than those in female professionals who did not report anxiety disorders.

In addition, the total effort expended in unpaid domestic work was significantly higher than that in female health care professionals without reports of anxiety disorders. Using the standardized effect size (SES), a moderate effect was found for the frustration domain. For the other significant differences, the effect was small.

Regarding male health care professionals, no difference was statistically significant. However, the standardized effect size (SES) in this group showed that the presence of anxiety disorders had a moderate effect on frustration levels in relation to unpaid domestic work, although this level was not statistically significant. The effect of anxiety disorders was stronger (SES) for men in relation to the total domestic workload.

An evaluation of the workload according to the presence or absence of episodes of depression according to the gender of the health care professionals is presented in Table 4 and Figure A5 (Appendix B). In female health care professionals who reported episodes of depression, the workload domains (mental demand, total effort, frustration, and total workload) in PHC work were significantly higher than those in female professionals who did not report episodes of depression. Furthermore, the workload domains (mental demand, physical demand, temporal demand, frustration, and total workload) in unpaid domestic work were significantly higher than those in women who did not report episodes of depression. From the standardized effect size (SES), a moderate effect was found for the frustration level in PHC work for female professionals that reported episodes of depression. For other significant differences, the effect was small.

In male health care professionals who self-reported episodes of depression, the level of frustration in relation to unpaid domestic work was significantly higher than in professionals who did not report episodes of depression. In addition, in unpaid domestic work, the effect size was large for the relationship between frustration level and the presence of episodes of depression. For the others, even without statistical significance, depressive episodes had a moderate effect on mental demand, temporal demand, performance and total workload in relation to the unpaid domestic workload. Again, the effect of episodes of depression was stronger in relation to unpaid domestic work for male health care professionals. Figure 2 presents a synthesis of the results that were evaluated as statistically significant according to the analysis of health workers, focusing on gender difference in terms of workload for PHC and unpaid domestic work. The figure presents the results of these workloads when considering their individual association with anxiety disorders and episodes of depression.

## 4. Discussion

The group of participants in the present study was predominantly composed of female health care professionals. This confirmed the social phenomenon of the feminization of health care professionals in PHC [4,5]. It was also observed that this group of women had been working in PHC for longer and completed fewer hours of work per week compared to the group of male health care professionals. This evidence indicates that this group of men completed a greater proportion of other work in parallel to PHC. This fact is socially justified in view of the social norms, which establish that the man should be the family representative and earn the majority of the family’s financial income. Another study showed that the nursing profession in a hospital environment is mostly composed of women; however, unlike the results of the present study, women had other parallel jobs in PHC [14]. This evidence allowed for us to conclude that the workers that participated in the study reproduced the more general context of PHC work in Brazil; that is, 78.9% of the workforce in the primary health care network in the country is female [27].

The present study revealed an association between workload scores, both for work in PHC and unpaid domestic work, and the gender of health professionals. Female professionals had higher workload scores than male professionals, with an emphasis on mental demand and total effort. These indications regarding women health professionals reinforce the magnitude of the mental workload, since the theory is that mental demand refers to the amount of mental resources needed to perform a set of simultaneous tasks and the total effort that is required. This alludes to the mental and physical difficulty that workers face when attempting to reach the level of performance that the activity demands [13]. It is important to underline that these conceptions emphasize the interpretation of the phenomenon of workload as an indicator of female vulnerability, considering the PH working conditions and the family environment as inducing illness in these professionals.

Consistent with the results of the present study, the high workloads in terms of mental and physical demands on the PHC team were confirmed in a study on psychosocial factors in health professionals; during the acute period of the COVID-19 pandemic, these were predominantly found in women, with a high mental workload being required to plan, execute, and maintain work activities [43]. It is worth noting that the study was carried out in an extreme situation, as in the case of the pandemic, in contrast to the results of the present research, which demonstrated, prior to this extreme and global situation, high scores regarding the physical and mental workload required in the PHC work. In addition, the physical and mental effort of these female healthcare providers mainly refers to the long working hours, number of night shifts, psychological demands related to patient care and low professional autonomy that lead to high levels of frustration [38,39]. This may present risks to the health of these female PHC workers, as it reflects the gender difference already established by the women’s susceptibility to anxiety problems and episodes of depression [44,45].

The mental demand for work in PHC can be explained by the need for health professionals to always be alert, as health services use PHC as a gateway to the health system that is responsible for most of the care provided to the patient population [46]. From this perspective, a study with PHC workers showed that precarious forms of work have consequences for workers’ health and their ability to assist the community, causing a direct decrease in the professionals’ productivity levels and, consequently, the quality of services provided by municipal health professionals of both sexes [47]. The identification of these working conditions can help to establish awareness and training actions for health professionals and PHC management, based on the gender divisions in work in PHC and unpaid domestic work, in a qualified and healthy way.

Female health professionals with high PHC workload scores also had a high unpaid domestic workload, above the levels presented by male professionals. An interesting aspect presented in more recent research highlights that the work developed in the job market, in this case, in primary health care, even if it provides satisfaction, parallels a greater degree of family interference in the work performed by women, which comes from their increased involvement in household chores, including taking care of the house, children and elderly relatives [48]. In addition, there is still a tendency for the work performed by women in PHC to lead to a greater susceptibility to gender inequities. This is because female professionals with a partner who also works full-time dedicate more time to unpaid domestic work compared to the male partner [48]. This can present risks to the health of these PHC workers, and reflects the gender differences that were previously established by women’s susceptibility to anxiety and episodes of depression.

According to European statistics, when considering the commuting time between home and work and unpaid domestic work, women work an average of 64 h per week compared to 53 h for men. Women spend an average of 26 h caring for children and elderly relatives, while men spend only 9 h [49]. It seems that, especially during parenting, women face a greater burden of unpaid domestic work due to family care [47]. This ideology is reflected in the social discourse regarding the traditional gender role model, in which the domain of paid work is shown to be more important for men, while the domain of unpaid domestic work is perceived to be more relevant for women [50].

As demonstrated, the potential for increased workload was directly associated with female health professionals, even after examining the current association between the gender of health professionals when determining the workload, especially in terms of mental and physical demands, total workload and frustration. Some studies in the literature corroborate this evidence and could work with the re-construction of a division of labor according to health workers’ gender [51,52] to increase workloads and the specificity of PHC workers.

Regarding the levels of frustration as a mechanism that induces workload, an explanation can be offered by a study carried out using different categories of health professionals, which showed that measures to reduce frustration with the workload can include reducing working hours, improving working conditions and pay, and the proper division of household and child-care tasks between men and women [53] The results of the present study demonstrate the need for future studies on the specificity of the relationship between well-being at work and productivity from a gender perspective.

Another highlight of the present study is the mental health of workers in a negative sense; that is, in the focus on anxiety and episodes of depression. The magnitude of the relationship with workers’ gender was confirmed, as these are more prevalent among women workers. Previous research carried out with nurses found similar results to this research and proved that social support from co-workers moderates the positive relationship between daily workload and episodes of depression, so that low levels of these are associated with reduced workload daily and sleep periods [54].

Continuing the evaluation of the workload and anxiety disorders according to the gender of the provider, the domains of mental workload, temporal, frustration, and total workload were higher in women with anxiety disorders than in those without inconveniences in PHC work and unpaid domestic work. Similar results were found in studies on health care workload, looking at the relationship between the gender of health professionals, PHC work and professionals that undertake unpaid domestic work, with samples mostly composed of women. Adverse working conditions, mainly related to the work environment, with insufficient work equipment, a large amount of time spent on administrative activities, unpredictable tasks, frequent interruptions to the performance of functions and excessive working hours, were found to be contributing factors to the level of frustration and anxiety disorders. This intensifies when workers feel pressured by daily tasks related to the house, their children, and spouse [54,55,56,57]. It is important to note that the work and family conflict phenomenon predominantly occurs among women, although the gender gap is small in European countries, for example [58].

Looking at the relationship between workload, episodes of depression and gender (women/men) showed that mental demand, overall/total workload, and the frustration domains of the PHC workload, and the mental demand, physical demand, temporal demand, frustration, and overall/total workload domains of the domestic workload were significantly higher in those without episodes of depression. According to the standardized effect size (SES), a medium effect was found in the frustration domain. The effect on other significant differences was small. Studies that determined the levels of stress, depression and burnout in nurses during the COVID-19 pandemic showed that female health care professionals, particularly nurses, are the most psychosocially affected by episodes of depression and exhaustion resulting from their workload, including patient care, occupational hazards in primary health care units, emergency departments, infection control units and intensive care units [59,60]. This demonstrates the high workload of this category of health workers when aiming to maintain the quality of the work provided and their commitment to the profession, even when faced with adverse situations.

Care should be paid to not naturalize the workload as being typical of the female gender, reducing its force as a precondition for illness. The relationship between gender and work is socially structured and often neglected, particularly the importance of family care work as a precondition of the high unpaid domestic workload that is often embedded as a female characteristic. The results also showed an association between the domestic workload, levels of frustration and the gender of professionals; however, this result was controversial regarding mental disorders.

It was not possible to confirm the association between workload and anxiety disorders in either work environment for male health professionals. It is worth noting that the presence of anxiety disorders had an average effect (SES) on levels of frustration in relation to unpaid housework. In other work environments, studies show that male health professionals, including doctors and scientific staff from universities and university hospitals, fall into this occupational group, with long working hours, irregular shifts, and unstable work conditions, contributing to the increased distance from unpaid domestic work. [61,62]. These social working conditions corroborate the production of family and work conflicts, which often lead to men being absent from work due to stress and anxiety. This explicit representation shows that the lowest occupational position does not have higher rates of anxiety disorders.

It is also important to highlight that, in this group of male health professionals with episodes of depression, the level of frustration in relation to unpaid domestic work was significantly higher than in those without episodes of depression. For this variable, the effect size (SES) was large. For other variables, even those without statistical significance, depressive episodes had a medium effect on the mental demands, temporal demands, performance and overall/total workload domains in relation to the domestic workload. Studies carried out in Finland had similar results to this study [63]. However, a study performed in Germany found that the workload, mainly referring to physical exertion, had a stronger effect on mental health and depressive symptoms in male health care professionals when compared to the domestic workload [64]. These episodes of depression may refer to occupational health and safety measures that do not consider the physical and psychosocial environment of the work, such as occupational hierarchies, as well as conflicts between the worker’s family requirements and needs.

Although the present study did not present a way to minimize the relationship between workload and mental disorders, the results present the urgent need to develop interdisciplinary protocols that would help to change the workload from being determinant of illness to improving the well-being of health care professionals. Accordingly, these professionals’ work environment could be improved by reducing the workload domains, minimizing the problems and conditions regarding family involvement as a product of the division of labor by gender.

From this perspective, the potential for physical and mental costs, especially for women, persist with advances in the division of labor between men and women in general society. It is important to mention a report on Latin America and the Caribbean [7], in which demands are still imposed on women concerning advances in the paid labor market while maintaining the accumulating unpaid domestic work with family involvement. These characteristics are still present in this small research context. In other words, the metrics of physical and mental workload according to workers’ subjective opinion could express the broader conditions of the indicators presented in this report, as follows: “…the tensions between work and family are generating high costs for women, for people who require care and also for the economic growth of countries, the smooth functioning of the labor market and the productivity of companies” [7] (ILO 2009; p. 9).

The working conditions of the health professionals who participated in the research represent the broader context of the world of work. The findings of the present study represent a sociocultural and economic challenge. Furthermore, actions focused on workers can help to minimize and overcome these working and living conditions. These findings indicate a problem rooted in the broader and conflicting social structure, as they occur in all professionals who are overloaded at work. This is a global public health problem, experienced by different occupational groups in various types of work, and is usually predominantly experienced by women, as in the present study. To enact immediate changes in the area of healthcare, educational practices for health professionals, in the sense of significant learning regarding self-care, could lead to healthy habits being encouraged in day-to-day work, and the mutuality of different family responsibilities. Furthermore, management services need to work together with these health professionals to agree on ways to organize and operationalize community care practices and strengthen health education to promote comprehensive health care for workers.

## 5. Limitations

This study has some limitations, which are as follows: the cross-sectional design of the study cannot establish a causal link between the investigated variables and does not allow for us to formulate conclusions over time, while convenience sampling does not allow for the results to be generalized to all health care professionals. Therefore, a longitudinal study with a randomized sample is suggested to provide a more in-depth study of the present topic. The evidence from the present study demonstrates the need to analyze the issues related to the health of primary health care workers. A productive and sustainable workforce requires a healthy and safe working environment, and for workers to positively respond to themselves and their family, while remaining active in the labor market [65]. This study reveals the workload produced by unpaid domestic work that health workers must undertake to maintain a healthy and harmonious environment for their family. Thus, in the context of this study, there is a need to intensify the integrated actions of public health services regarding their workers, with a focus on primary health care (PHC) workers.

## 6. Conclusions

The workers in the present study perceive there to be human costs, especially female health professionals, facing work in primary health care (PHC) and family responsibilities due to the division of labor by gender. PHC workload and unpaid domestic workload were higher in women with anxiety disorders and episodes of depression. For men who experienced episodes of depression, domestic workload had a large effect on the level of frustration with family involvement.

Even considering the limitations of the study, the workload metric was identified as an important indicator of female vulnerability in relation to the working conditions in PHC and the family environment.

Based on this knowledge, it is recommended that interventions are planned with interdisciplinary teams and PHC workers and managers to strengthen the policies aimed at minimizing the working conditions that lead to work overloads. Furthermore, the development of longitudinal studies with different samples of health professionals would be relevant to highlight the differences between men and women regarding the division of labor by gender.

## Figures and Tables

**Figure 1 ijerph-19-09816-f001:**
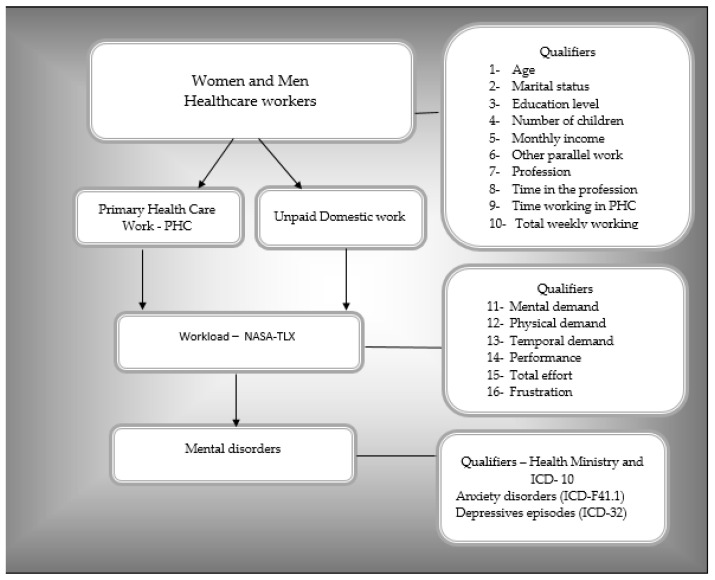
Outline of the main concepts used (PHC workload and unpaid domestic work; mental disorders (anxiety disorders and episodes of depression)) and their qualifiers to investigate the magnitude of the problem among female and male primary health care professionals.

**Figure 2 ijerph-19-09816-f002:**
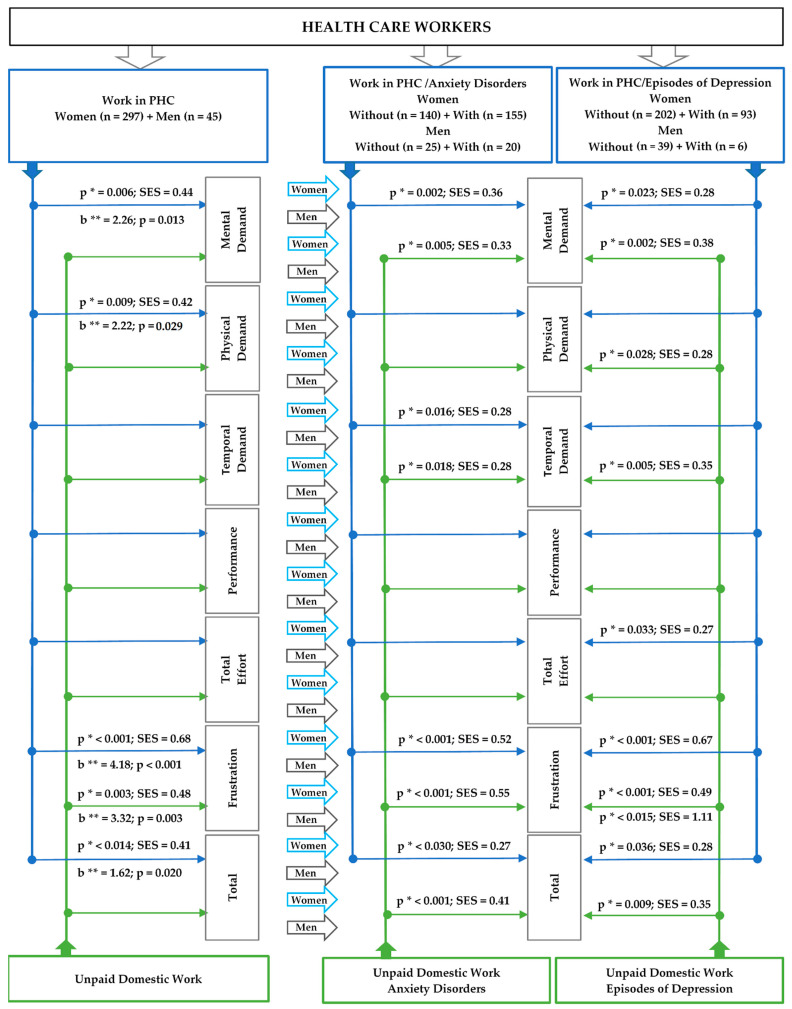
Synthesis of the results evaluated as statistically significant from the analysis by gender in terms of workload in PHC and the results of this workload when considering, one at a time, the association with anxiety disorders and with episodes of depression. * Test *t*-student; ** regression coefficient (95% confidence interval) adjusted for age, marital status, education level, number of children, monthly income, other parallel work, profession, time in the profession, time working in PHC and total weekly working hours; SES: standardized effect size (Cohen).

**Table 1 ijerph-19-09816-t001:** Characterization of the total sample according to the gender of the health care professionals.

Variables	Total Sample (*n* = 342)	Men (*n* = 45)	Women (*n* = 297)	*p*
	*n* (%)	*n* (%)	*n* (%)	
**Age (years) ***	41.5 ± 10.0	38.6 ± 11.8	42.0 ± 9.7	0.033 ^a^
**Skin color**				0.137 ^b^
Caucasian	262 (77.5)	31 (70.5)	231 (78.6)	
African descendant	37 (10.9)	4 (9.1)	33 (11.2)	
Mixed race	39 (11.5)	9 (20.5)	30 (10.2)	
**Marital status**				0.093 ^b^
Single	99 (28.9)	20 (44.4)	79 (26.6)	
Married/stable union	200 (58.5)	21 (46.7)	179 (60.3)	
Separated/divorced	37 (10.8)	3 (6.7)	34 (11.4)	
Widowed	6 (1.8)	1 (2.2)	5 (1.7)	
**Level of education**				0.002 ^b^
Incomplete high school/complete high school	83 (24.3)	4 (8.9)	79 (26.6) ^#^	
Technical course	47 (13.7)	3 (6.7)	44 (14.8)	
Incomplete higher education	39 (11.4)	5 (11.1)	34 (11.4)	
Complete higher education/technology course	98 (28.7)	24 (53.3) ^#^	74 (24.9)	
Specialization	59 (17.3)	6 (13.3)	53 (17.8)	
Master’s/doctorate	16 (4.7)	3 (6.7)	13 (4.4)	
**Number of children ****	1 (0–2)	0 (0–2)	1 (1–2)	0.002 ^c^
**Monthly income (m.w.)**				<0.001 ^b^
Up to 2	130 (38.7)	11 (25.0)	119 (40.8) ^#^	
From 2 to 4	116 (34.5)	10 (22.7)	106 (36.3)	
From 4 to 8	41 (12.2)	6 (13.6)	35 (12.0)	
More than 8	49 (14.6)	17 (38.6) ^#^	32 (11.0)	
**Location**				0.998 ^b^
Municipality 1	286 (83.6)	38 (84.4)	248 (83.5)	
Municipality 2	56 (16.4)	7 (15.6)	49 (16.5)	
**Other work parallel to that of PHC**				<0.001 ^b^
Yes	63 (18.5)	19 (43.2)	44 (14.9)	
No	277 (81.5)	25 (56.8)	252 (85.1)	
**Sector-PHU**				0.918 ^b^
PHU—traditional	26 (7.6)	4 (8.9)	22 (7.4)	
FHU	282 (82.5)	37 (82.2)	245 (82.5)	
PHU—24 h	34 (9.9)	4 (8.9)	30 (10.1)	
**Profession**				<0.001 ^b^
Nurse	53 (15.5)	2 (4.4)	51 (17.2) ^#^	
Doctor	43 (12.6)	18 (40.0) ^#^	25 (8.4)	
Nursing technician/assistant	73 (21.3)	9 (20.0)	64 (21.5)	
Community health agent	139 (40.6)	12 (26.7)	127 (42.8) ^#^	
Dentist	13 (3.8)	4 (8.9)	9 (3.0)	
Oral health technician/assistant	15 (4.4)	0 (0.0)	15 (5.1)	
Other	6 (1.8)	0 (0.0)	6 (2.0)	
**Length of time in the profession (years) ****	11 (4–16)	5 (0–16)	11.5 (7–16)	0.011 ^c^
**Time working in PHC (years) ****	8 (1–13)	2 (0–9)	9 (2–14)	<0.001 ^c^
**Weekly workload**	44.4 ± 13.2	48.3 ± 13.8	43.8 ± 13.0	0.033 ^a^
**Work period in PHC**				0.916 ^b^
Day shift	304 (89.4)	41 (91.1)	263 (89.2)	
Night shift	8 (2.4)	1 (2.2)	7 (2.4)	
Night/day	28 (8.2)	3 (6.7)	25 (8.5)	

^a^ Student’s *t*-test; ^b^ Pearson’s chi-square test; ^c^ Mann–Whitney test; * described by mean ± *SD*; ** described by median (25th–75th percentiles); ^#^ statistically significant association by the residuals test adjusted to 5% significance.

**Table 2 ijerph-19-09816-t002:** Comparison of the gender of health care professionals in terms of PHC workload and unpaid domestic work.

Workload Domains	Men (*n* = 45)	Women (*n* = 297)	*p* *	Cohen’s SES	*b* (95%CI) **	*p*
	Mean ± *SD*	Mean ± *SD*				
**Workload in PHC work**						
Mental demand	13.5 ± 5.9	15.8 ± 5.1	0.006	0.44	2.26 (0.47 to 4.04)	0.013
Physical demand	10.5 ± 5.1	12.9 ± 5.6	0.009	0.42	2.22 (0.23 to 4.20)	0.029
Temporal demand	12.9 ± 5.4	14.4 ± 5.4	0.089	0.28	1.81 (−0.08 to 3.71)	0.061
Performance	13.7 ± 5.1	14.6 ± 5.4	0.303	0.17	0.88 (−1.00 to 2.77)	0.359
Total effort	13.9 ± 4.7	15.5 ± 4.6	0.024	0.36	1.54 (−0.11 to 3.20)	0.067
Frustration	8.8 ± 5.5	12.8 ± 6.0	<0.001	0.68	4.18 (2.17 to 6.19)	<0.001
Total	13.6 ± 4.1	15.1 ± 3.7	0.014	0.41	1.62 (0.25 to 2.98)	0.020
**Workload in unpaid domestic work**						
Mental demand	12.1 ± 5.8	13.2 ± 6.0	0.241	0.19	1.28 (−0.88 to 3.43)	0.244
Physical demand	9.2 ± 4.9	11.5 ± 5.8	0.007	0.39	1.95 (−0.08 to 3.98)	0.059
Temporal demand	12.3 ± 4.7	13.5 ± 5.5	0.148	0.24	1.34 (−0.65 to 3.33)	0.187
Performance	13.0 ± 5.4	13.0 ± 5.4	0.958	0.01	0.15 (−1.79 to 2.09)	0.878
Total effort	12.6 ± 5.5	14.2 ± 4.9	0.050	0.32	1.03 (−0.76 to 2.82)	0.259
Frustration	7.3 ± 5.6	10.2 ± 6.2	0.003	0.48	3.32 (1.12 to 5.52)	0.003
Total	12.0 ± 4.2	13.4 ± 4.2	0.049	0.33	1.42 (−0.13 to 2.97)	0.073

* Student’s *t*-test; ** regression coefficient (95% confidence interval) adjusted for age, marital status, education level, number of children, monthly income, parallel work, profession, time in the profession, time working in PHC and total weekly working hours; SES: standardized effect size.

**Table 3 ijerph-19-09816-t003:** Comparison according to the gender of the health care professionals in terms of PHC workload and unpaid domestic work according to the presence or absence of anxiety disorders.

	Women (*n* = 295)				Men (*n* = 45)			
Workload Domains	Without Anxiety Disorders (*n* = 140)	With Anxiety Disorders (*n* = 155)	*p* *	Cohen’s SES	Without Anxiety Disorders (*n* = 25)	With Anxiety Disorders (*n* = 20)	*p* *	Cohen’s SES
	Mean ± *SD*	Mean ± *SD*			Mean ± *SD*	Mean ± *SD*		
**Workload in PHC work**								
Mental demand	14.8 ± 5.3	16.6 ± 4.7	0.002	0.36	13.1 ± 5.5	13.9 ± 6.5	0.644	0.14
Physical demand	12.5 ± 5.9	13.2 ± 5.3	0.256	0.13	11.0 ± 5.5	10.0 ± 4.6	0.536	0.19
Temporal demand	13.6 ± 5.4	15.1 ± 5.2	0.016	0.28	13.3 ± 5.0	12.5 ± 5.9	0.630	0.15
Performance	14.6 ± 5.2	14.6 ± 5.5	0.889	0.02	13.4 ± 5.5	14.2 ± 4.7	0.614	0.15
Total effort	15.0 ± 4.6	16.0 ± 4.6	0.062	0.22	13.6 ± 4.4	14.3 ± 5.1	0.629	0.15
Frustration	11.2 ± 6.0	14.2 ± 5.6	<0.001	0.52	9.3 ± 5.7	8.2 ± 5.2	0.496	0.21
Total	14.6 ± 3.8	15.6 ± 3.5	0.030	0.27	13.5 ± 4.2	13.8 ± 4.1	0.819	0.07
**Workload in unpaid domestic work**								
Mental demand	12.2 ± 5.9	14.1 ± 6.0	0.005	0.33	10.9 ± 5.5	13.6 ± 6.0	0.134	0.46
Physical demand	10.8 ± 5.8	12.1 ± 5.8	0.052	0.23	10.1 ± 5.0	8.2 ± 4.6	0.201	0.39
Temporal demand	12.7 ± 5.4	14.3 ± 5.5	0.018	0.28	11.4 ± 4.5	13.5 ± 4.9	0.142	0.46
Performance	13.3 ± 5.2	12.8 ± 5.6	0.456	0.09	13.3 ± 4.7	12.6 ± 6.3	0.682	0.12
Total effort	13.6 ± 4.9	14.7 ± 4.9	0.043	0.24	12.1 ± 5.1	13.3 ± 6.0	0.500	0.20
Frustration	8.5 ± 5.5	11.7 ± 6.4	<0.001	0.55	6.0 ± 4.8	8.8 ± 6.2	0.101	0.50
Total	12.5 ± 4.1	14.2 ± 4.2	<0.001	0.41	11.4 ± 4.1	12.8 ± 4.3	0.286	0.33

* Student’s *t*-test; SES: standardized effect size.

**Table 4 ijerph-19-09816-t004:** Comparison by gender of the health care professionals in terms of PHC workload and unpaid domestic work, according to the presence or absence of episodes of depression.

	Women (*n* = 295)				Men (*n* = 45)			
Workload Domains	Without Episodes of Depression (*n* = 202)	With Episodes of Depression (*n* = 93)	*p* *	Cohen’s SES	Without Episodes of Depression (*n* = 39)	With Episodes of Depression (*n* = 6)	*p* *	Cohen’s SES
	Mean ± *SD*	Mean ± *SD*			Mean ± *SD*	Mean ± *SD*		
**Workload in PHC work**								
Mental demand	15.3 ± 5.2	16.7 ± 4.6	0.023	0.28	13.4 ± 5.9	14.2 ± 6.6	0.766	0.13
Physical demand	12.8 ± 5.9	13.0 ± 5.1	0.767	0.04	10.7 ± 5.4	9.5 ± 2.3	0.370	0.23
Temporal demand	14.1 ± 5.3	15.0 ± 5.4	0.165	0.18	13.2 ± 5.4	11.2 ± 5.6	0.407	0.37
Performance	14.6 ± 5.2	14.6 ± 5.8	0.986	0.00	13.4 ± 5.3	15.7 ± 3.2	0.323	0.44
Total effort	15.2 ± 4.6	16.4 ± 4.5	0.033	0.27	14.1 ± 4.8	12.5 ± 3.8	0.449	0.34
Frustration	11.6 ± 5.7	15.4 ± 5.7	<0.001	0.67	9.0 ± 5.6	7.5 ± 4.7	0.544	0.27
Total	14.8 ± 3.7	15.8 ± 3.5	0.036	0.28	13.5 ± 4.3	13.9 ± 2.9	0.840	0.09
**Workload in unpaid domestic work**								
Mental demand	12.5 ± 5.8	14.8 ± 6.2	0.002	0.38	11.5 ± 5.7	15.8 ± 5.6	0.091	0.76
Physical demand	11.0 ± 5.8	12.6 ± 5.8	0.028	0.28	9.3 ± 4.6	9.2 ± 7.1	0.967	0.02
Temporal demand	12.9 ± 5.4	14.9 ± 5.7	0.005	0.35	11.8 ± 4.5	15.2 ± 5.5	0.106	0.73
Performance	13.2 ± 5.2	12.7 ± 5.9	0.536	0.08	12.6 ± 5.4	15.5 ± 5.5	0.227	0.54
Total effort	13.8 ± 4.9	15.0 ± 4.9	0.071	0.23	12.7 ± 5.2	11.8 ± 7.7	0.710	0.16
Frustration	9.3 ± 5.9	12.2 ± 6.4	<0.001	0.49	6.5 ± 5.1	12.3 ± 6.4	0.015	1.11
Total	12.9 ± 4.1	14.4 ± 4.3	0.009	0.35	11.6 ± 4.1	14.3 ± 4.7	0.156	0.64

* Student’s *t*-test; SES: Cohen’s standardized effect size.

## Data Availability

Data regarding this study can be shared upon request to the corresponding author. The data are not publicly available due to ethical issues.

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
