# Peer review of "Domains of Physical and Mental Workload in Health Work and Unpaid Domestic Work by Gender Division: A Study with Primary Health Care Workers in Brazil"

_ijerph, 2022, doi:10.3390/ijerph19169816_

Round 1

Reviewer 1 Report

In general, this manuscript is well described. Although the authors followed a rigorous process to finally present it, the described results and methodology do not go beyond the problem. 

The authors merely stay on an inferential description of the sample. It is true that this is important,  but the journal you are submitting the paper, usually looks for an additional perspective. 

Some target comments:

Introduction: please state clearly what you mean by "workload". I did not understand what the article brings to the existing theory. You emphasised some limitations on the existing theories,  but you are not offering new opportunity to face them. Kindly please review this section thinking about your real objective is, and why it could be important for other research in the same field.

Method: please state some information on the instruments fiability (total and per dimensions). Also inform on the average time of completion.

If possible include the spss syntax in order to enhance reproducibility. 

Author Response

Dear Reviewer,

We are writing to provide peer-to-peer responses to your comments. We carried out the recommendations and made improvements to the manuscript.

Comments and Suggestions for Authors

In general, this manuscript is well described. Although the authors followed a rigorous process to finally present it, the described results and methodology do not go beyond the problem. 

The authors merely stay on an inferential description of the sample. It is true that this is important, but the journal you are submitting the paper, usually looks for an additional perspective. 

Response:

The authors corroborate with the limits of the methodological design and its consequences, however, the study in its theoretical-scientific consistency allowed us to present a complex problem that integrates the concepts of workload, professional and domestic, from a gender perspective, so that the metrification to express the content in an integrated way, since studies in the literature on the workload of PHC professionals and their unpaid domestic work were presented separately, that is, in the focus of one or another type of work, mostly in paid work, and not always related to the division of labor by gender.

Point 1: Introduction: Please state clearly what you mean by "workload". I didn't understand what the article brings to the existing theory. You have emphasized some limitations of existing theories, but you are not offering new opportunities to address them. Please review this section with your real objective in mind and why this might be important for other research in the same field.

Response 1: The theoretical concept of workload that guided the study was improved in the introduction in paragraph 5 between lines 90-107, and other theoretical and operational elements of the metric were added to the method in paragraphs 8 and 9, between lines 287 - 297 and 298-315, respectively. (Highlighted in red in the manuscript text)

It is emphasized that the text brings to the existing theory and to the field of research on the workload elements of the integration between two concepts of work, that is, professional work and unpaid domestic work against the concept of the division of labor by gender, as presented in paragraph 15. “It is important to highlight that studies in the literature on the workload of PHC workers and unpaid domestic work were presented separately, that is, focusing on one or another type of work, mostly on paid work, and neither always related to the division of labor by gender. In this sense, it is worth highlighting the present study, which integrates the study of workload in two work environments, work in PHC and unpaid domestic work, from the perspective of the division of labor by gender. In view of this finding, comprehensive measures and public health actions are imperative, which require adjustments in the model of worker health care in PHC, with local and global political decisions that take into account the assessment of the workload accumulation. The workload generated in these two work environments can affect the health professional, that is, the workload in PHC can affect unpaid domestic work due to its socially complex nature that contributes to work-family conflict and illness due to mental disorders. Thus, this study becomes relevant for the integration of the university as a promotion of research and PHC management services for the development of management reports that promote the implementation of integrated projects and programs on the workload by gender division in the work environments, with the aim of promoting health in the biopsychosocial fields, well-being and strengthening the rights of this workforce, as well as subsidizing further research in the same field.

Point 2: Method: provide some information about the reliability of the instruments (total and by dimensions). Also enter the average completion time.

Response 2: In the Method, information about the reliability of the instruments was included, in addition to other details of the NASA-TLX scale throughout the methodological construction, targeted highlights: -

- Added the NASA-TLX scale model - Figure 3 – Appendix A1 (p.19)- The NASA-TLX scale was applied to health care workers twice, once corresponding to their experience in primary health care work and another about their domestic work with their family;

- Redoing Cronbach's alpha, the values ​​of .805 for the domestic workload (unpaid domestic work) and .767 for the workload in primary health care (PHC) work were maintained. According to the scale structure, Cronbach's alpha is calculated covering the set of scale dimensions.

The average time was 58 minutes for the individual and face-to-face interview with health workers. It should be noted that the application of the scale was included in the interview and the guidelines were maintained during this process.

Point 3: If possible, include the spss syntax to increase reproducibility.

Response 3: As suggested, we have included the SPSS Syntax program in PDF format, as an attachment. Please see the attachment.

Conclusions have been improved.

We appreciate all comments made. Thanks very much.

Reviewer 2 Report

Overall, the purpose, methods, and results of the study were consistent, and the background and discussion were well described.

I recommend adding the definition of the tool in the measurement section.

Author Response

Dear Reviewer,

We are writing to provide peer-to-peer responses to your comments. We carried out the recommendations and made improvements to the manuscript.

Comments and suggestions for authors

Overall, the purpose, methods, and results of the study were consistent, and the background and discussion were well described.

Point 1: I recommend adding the definition of the tool in the measurement section.

Response 1: As recommended, the tool definition was added in the measurement section.

 - To identify the workload in the PHC work and unpaid domestic work, was applied the National Aeronautics and Space Administration (NASA) Task Load Index (TLX) (NASA-TLX) scale. It is designed to capture the subjective experience of workers [12-13]. This scale has already been applied in different work environments [30-34], and it has been validated in its conceptual and operational structures Brazilian studies [27-28]. Furthermore, the research group has used this scale in previous studies [35-40]. These studies prove that the operator/worker-centric NASA-TLX scale has consistency, sensitivity, and acceptance among operators. As well, as it is a multidimensional scale, it can be applied to obtain more detailed and diagnostic data that lead to the questioning of what is producing the workload reported by workers and can point out strategies for relieving excessive workload. (paragraph 8, p. 6-7);

 - Following NASA-TLX procedures, workload assessment results using this scale are classified into four workload levels as follows: 0–5: low; 6–10: medium-low; 11–15: medium-high; 16-20: high.

Added the NASA-TLX scale model - Figure 3 – Appendix A1 (p.19)- The NASA-TLX scale was applied to health care workers twice, once corresponding to their experience in primary health care work and another about their domestic work with their family. (paragraph 9, line 317-320, p. 7);

Furthermore, the concept of workload theory that guided the study was improved in the introduction in paragraph 5 between lines 90-107.

The health care worker was asked to assign a score from zero to 20 (zero = no demand and 20 = maximum demand) for workload in the six previously mentioned domains [13]. Following NASA-TLX procedures, workload assessment results using this scale are classified into four workload levels as follows: 0–5: low; 6–10: medium-low; 11–15: medium-high; 16-20: high. (Figure 3 – Appendix A1). The NASA-TLX scale was applied to health care workers twice, once corresponding to their experience in primary health care work and another about their domestic work with their family. The reliability of these variables was assessed through Cronbach's alpha, which was .805 for the domestic workload (unpaid domestic work) and .767 for the workload in primary health care (PHC) work.

We appreciate all comments made.

Reviewer 3 Report

Thank you for submitting your manuscript on the physical and mental workload domains and family involvement by gender division for publication consideration.

I appreciate your thorough review of literature and careful attention to the impact of gender on workload in primary health care settings in Grande do Sul, Brazil.

You describe the study process and research design, acknowledging its limitations. In terms of the review of literature, did you find any studies that correlated physical and mental workload and domestic work with actual differences in patient outcomes or employee performance outcomes?

Could you expand your discussion of the implications of your findings to practice, education, and research? (Lines 532-539). How might the workload domains be "reduced"?

What are some best practices around balancing the physical and mental workload to improve health outcomes among employees?

Do you have clear theoretical and operational definitions for your variables and constructs? For example, what is primary health care?

Are you using the WHO definition for this term? Were employees recruited from hospitals or only outpatient facilities? 

The concepts of "Unpaid domestic work" and "family involvement" seem to be synonymous. Could someone be engaged in unpaid domestic work that is unrelated to their family? (for example, assisting families in need in one's church or social support network).

Would the relatively small number of men in the sample (compared with women) be a limitation of your study? 

Thank you again for your research in this area and for considering this journal as a publication venue.

Author Response

Dear Reviewer,

We are writing to provide peer-to-peer responses to your comments. We carried out the recommendations and made improvements to the manuscript.

Comments and Suggestions for Authors

Thank you for submitting your manuscript on the physical and mental workload domains and family involvement by gender division for publication consideration.

I appreciate your thorough review of literature and careful attention to the impact of gender on workload in primary health care settings in Grande do Sul, Brazil.

 Point 1: You describe the study process and research design, acknowledging its limitations. In terms of the review of literature, did you find any studies that correlated physical and mental workload and domestic work with actual differences in patient outcomes or employee performance outcomes?

Response 1: According to a literature review, no study was found that associated the physical and mental workload in primary health care work and in unpaid domestic work in an integrated way, from a gender perspective that evidenced results with real differences, but those studies carried out in one or another work environment from a gender perspective were included throughout the manuscript.

Point 2: Could you expand your discussion of the implications of your findings to practice, education, and research? (Lines 532-539). How might the workload domains be "reduced"?

Response 2: It is reiterated that the study did not directly present results consistent with the implications for practice, education and research in order to establish a dialogue with the literature, but reflective elements were improved throughout the discussion to approach the request regarding the implications for practice, education and research.

Point 3: What are some best practices around balancing the physical and mental workload to improve health outcomes among employees?

Response 3: This research is part of a worker's health program in primary health care, linked to the Laboratory for the Study of Socioenvironmental Processes and Collective Health Production at the Federal University of Rio Grande, which involves undergraduate and postgraduate training, university extension and multidisciplinary research. In this academic context, it is worth justifying that the interventions unfolded in the present study for the workers involved in the research have not yet been carried out. It is important to note that the COVID-19 pandemic led these professionals to intense involvement in their work process and restrictions had to be respected (2020-2021years). It is believed, however, that the research group's proposal to carry out collective integrated actions will have the adherence of professionals in order to collaborate in this "post-pandemic" period. It should be noted that the practical activities programmed by the research group are guided by the guidelines of the WHO and the Brazilian Ministry of Health on “care work and caregivers for a future with decent work” within the model of a healthy work environment”.

Point 4: Do you have clear theoretical and operational definitions for your variables and constructs? For example, what is primary health care?

Response 4: The theoretical definitions guiding the study can be found in the introduction and in the method. These consist of workload, work in primary health care (PHS), unpaid domestic work and gender division of labor. Furthermore, as recommended, the definition of primary health care was added to the introduction, as well as other defining elements on workload and unpaid domestic work. In the method, in addition to sociodemographic and occupational variables, work in PHC and the presence/absence of anxiety disorders and episodes of depression, other details were added about the metric to assess the subjective workload in PHC work and unpaid domestic work, the scale of NASA- TLX (National Aeronautics and Space Administration (NASA) Task Load Index - TLX) [12-13].

Point 5: Are you using the WHO definition for this term? Were employees recruited from hospitals or only outpatient facilities? 

Response 5: The WHO definition for the term - Primary Health Care (PHC) and its guiding and operational structure of the Brazilian Ministry of Health was used. Thus, the recommendation was met in the explanation of the term in the introduction.

Point 6: The concepts of "Unpaid domestic work" and "family involvement" seem to be synonymous. Could someone be engaged in unpaid domestic work that is unrelated to their family? (for example, assisting families in need in one's church or social support network).

Response 6: The authors corroborate the indication that the concepts of "unpaid domestic work" and "family involvement" are synonymous. Thus, the conceptual nomenclature "unpaid domestic work" was adopted, as described in the introduction.

Point 7: Would the relatively small number of men in the sample (compared with) be a limitation of your study?

Response 7: The small number of men in the sample (compared to women) in the context of the study, represents the composition of the professionals of the primary care network in outpatient health, and corresponds to the reality of this constitution throughout the Brazilian territory. It is characteristic was better detailed in the Method.

Figure 2 was inserted in the results section with the synthesis of the results evaluated as statistically significant from the analysis by gender of health workers in terms of workload at work in the PHC and unpaid domestic work; and the results of these workloads when considering, one at a time, the association with anxiety disorders and with episodes of depression.

We appreciate all comments made.

Round 2

Reviewer 1 Report

I find that the revised document has improved over its previous version

I still have the same concerns regarding the contribution of the document to the journal, even so I will change my opinion to "accepted", for two reasons: 1) because it seems important to me to make the topic they are dealing with visible; 2) taking into account the verdict of the other reviewers, and that the journal editor considers the document appropriate for the scope.

I invite the authors to raise their methods further for future research.